# The outcomes of SGLT-2 inhibitor utilization in diabetic kidney transplant recipients

Jia-Yuh Sheu [1], Li-Yang Chang[2], Jui-Yi Chen [3,4], Heng-Chih Pan[5,6,7,8], Chi-Shin Tseng [1,9], Jeff S. Chueh [1,2,9,12] ✉ & Vin-Cent Wu [9,10,11,12] ✉

Sodium-glucose cotransporter 2 inhibitors (SGLT-2i) have demonstrated efficacy in reducing cardiovascular events and potentially improving kidney function in diabetic patients. This investigation analyzes the TriNetX database to assess the efficacy of SGLT-2i in diabetic kidney transplant recipients (KTR) concerning all-cause mortality, major adverse cardiac events (MACE), and major adverse kidney events (MAKE). The study includes type 2 diabetic patients over 18 who underwent kidney transplants between June 1, 2015, and June 1, 2023, with a focus on SGLT-2i use within the first three months post-transplant. After propensity score matching, the study compares 1970 SGLT-2i users with matched non-users. With a median follow-up of 3.4 years, SGLT-2i users showed significantly lower rates of all-cause mortality (adjusted hazard ratio [aHR]: 0.32), MACE (aHR: 0.48), and MAKE (aHR: 0.52). These findings indicate that SGLT-2i significantly reduces mortality and adverse events in diabetic KTR, underscoring its potential to improve post-transplant outcomes.

The escalating incidence of diabetes mellitus leading to end-stage kidney disease (ESKD) and necessitating kidney replacement therapy has become a prominent concern[1,2]. Diabetes mellitus control emerges as a significant challenge following kidney transplantation. This condition is associated with a heightened risk of early-onset cardiovascular disease and increased mortality rates among kidney transplant recipients (KTR)[3,4]. Sodium-glucose cotransporter 2 inhibitors (SGLT-2i) have demonstrated efficacy in mitigating cardiovascular events and potentially ameliorating kidney function[5–8]. In individuals with type 2 diabetes and concurrent kidney disease, SGLT-2i are linked to a reduced risk of kidney failure and cardiovascular complications[9,10].

Despite these advancements, the efficacy of SGLT-2i among diabetic patients who have undergone kidney transplantation remains less well elucidated. A Korean study suggests that SGLT-2i therapy can be safely and effectively employed in diabetic KTR to preserve graft function[11]. Another observational study revealed that SGLT-2i could enhance blood sugar control, reduce body weight, lower blood pressure, and positively influence other health markers[12]. Consequently, our research endeavors to fill this knowledge gap by examining the impact of SGLT-2i on the outcomes of diabetic KTR. TriNetX, a global collaborative health research network platform, facilitates access to electronic medical records and data from diverse healthcare organizations, reflecting real-world practices. Real-world data holds immense potential for designing and executing confirmatory trials, addressing questions that might otherwise remain unexplored[13].

In this work, we leveraged the international TriNetX platform to investigate whether the use of SGLT-2i can reduce all-cause mortality, major adverse cardiac events (MACE), and major adverse kidney events (MAKE) in diabetic KTR. Our findings demonstrate that SGLT-2i use is associated with significantly lower rates of these adverse

[1]Department of Urology, National Taiwan University Hospital, Taipei, Taiwan. [2]College of Medicine, National Taiwan University, Taipei, Taiwan. [3]Division of Nephrology, Department of Internal Medicine, Chi-Mei Medical Center, Tainan, Taiwan. [4]Department of Health and Nutrition, Chia Nan University of Pharmacy and Science, Tainan, Taiwan. [5]Graduate Institute of Clinical Medicine, College of Medicine, National Taiwan University, Taipei, Taiwan. [6]Chang Gung University College of Medicine, Taoyuan, Taiwan. [7]Division of Nephrology, Department of Internal Medicine, Keelung Chang Gung Memorial Hospital, Keelung, Taiwan. [8]Community Medicine Research Center, Keelung Chang Gung Memorial Hospital, Keelung, Taiwan. [9]Primary Aldosteronism Center, National Taiwan University Hospital, Taipei, Taiwan. [10]Division of Nephrology, Department of Internal Medicine, National Taiwan University Hospital, Taipei, Taiwan. [11]National Taiwan University Hospital Study Group of Acute Renal Failure and Taiwan Consortium for Acute Kidney Injury and Renal Diseases, Taipei, Taiwan. [12]These authors contributed equally: Jeff S. Chueh, Vin-Cent Wu. ✉e-mail: JeffChueh@gmail.com; q91421028@ntu.edu.tw

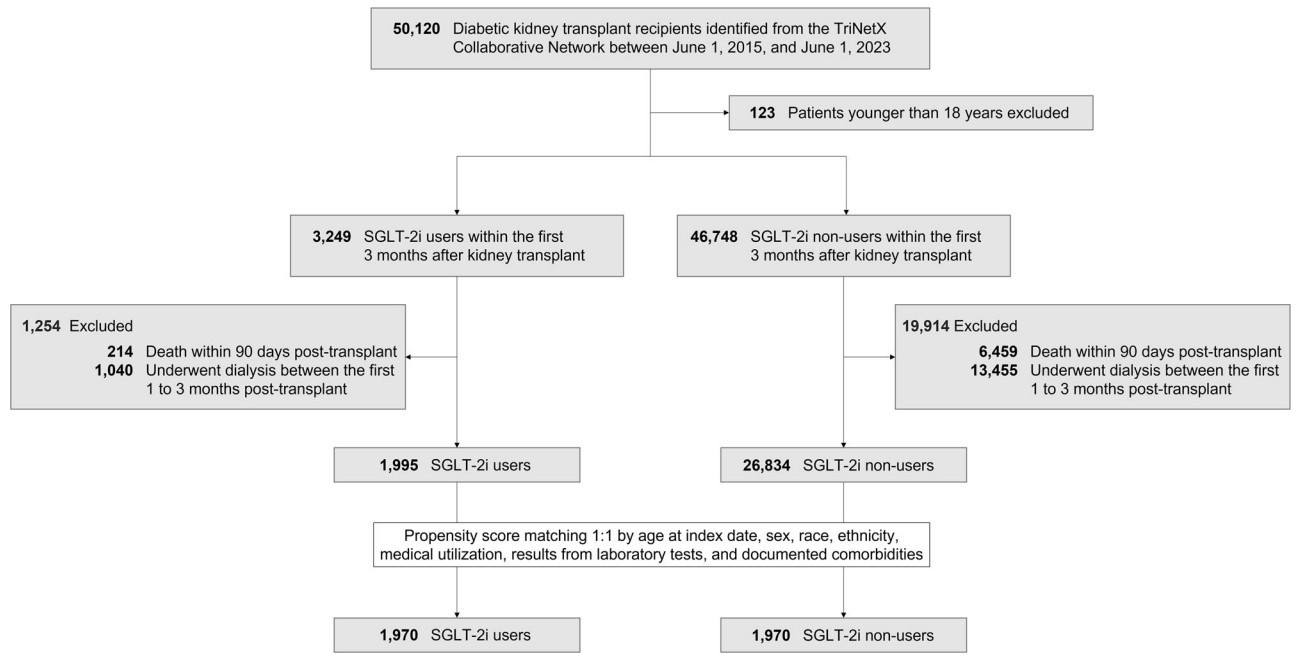

**Fig. 1 | Algorithm for patient selection and enrollment in the study.** SGLT-2i, sodium-glucose cotransporter 2 inhibitor.

outcomes, suggesting its potential to improve long-term survival and reduce complications in this patient population.

## Results

### Study Population Characteristics

In our comprehensive study, we analyzed 50,120 diabetic KTR (Fig. 1). This led to a refined cohort comprising 3249 (6.5%) individuals on SGLT-2i and 46,748 who were not, after excluding 123 patients who were less than 18 years of age. We further refined our analysis by identifying 1995 diabetic KTR with SGLT-2i users who neither passed away nor required dialysis within 1 to 3 months post-transplant. A comparable group of 26,834 non-users was also selected. Among these groups, 66 (3.3%) of the 1,995 SGLT-2i users were diagnosed with post-transplant diabetes mellitus (PTDM), compared to 2218 (8.3%) of the 26,834 nonusers. The SGLT-2i users were older than the SGLT-2i non-users (mean age $59.5 \pm 11.3$ years vs. $57.2 \pm 12.3$ years). There was a higher proportion of male in the SGLT-2i group compared to the SGLT-2i non-users (63.4% vs. 58.6%, standardized difference = 0.10). The proportion of individuals of White ethnicity was higher in the SGLT-2i non-users group compared to the SGLT-2i users. Despite these distinctions, both groups exhibited similarities in most demographic, comorbidities, and other pertinent factors. To facilitate a comprehensive and equitable comparison, we employed propensity score matching (PSM) to create two meticulously balanced groups: 1970 SGLT-2i users and an equivalent number of non-users. Following the PSM procedure, these groups demonstrated close alignment in terms of age, sex, race, ethnicity, comorbidities, medication use, and the majority of laboratory results (Table 1).

### SGLT-2i usage and outcomes in diabetic KTR

After a median follow-up of 3.4 years, the mortality rate among SGLT-2i users was significantly lower, with 2.08% (41 out of 1970, 6.56 per 1000 person-years) patients experiencing mortality, compared to 9.54% (188 out of 1970, 33.05 per 1000 person-years) in the non-user group (Supplemental Table S1 & S2). This corresponds to a risk difference of -7.46% (95% confidence interval [CI]: -8.90% to -6.02%, $p < 0.001$), indicating a significant reduction in absolute risk. The risk ratio was 0.22 (95% CI: 0.16–0.30), and the odds ratio was 0.20 (95% CI:

0.14–0.28) (Supplemental Table S3). The adjusted hazard ratio (aHR) for all-cause mortality among SGLT-2i users was notably reduced at 0.32 (95% CI: 0.22–0.45, $p < 0.001$) compared to non-users (Fig. 2). Furthermore, the incidence of MACE was substantially lower in the SGLT-2i user group (4.44%, 72 out of 1623) than in the non-user group (13.87%, 229 out of 1651), with an aHR of 0.48 (95% CI: 0.37 - 0.62, $p < 0.001$). Additionally, the occurrence of MAKE in SGLT-2i users (8.93%, 176 out of 1970) was considerably less than in non-users (22.54%, 444 out of 1970), leading to an aHR of 0.52 (95% CI: 0.43–0.62, $p < 0.001$). Supplemental Fig. S1 presents the Kaplan-Meier Survival Curves for our primary and secondary outcomes. Landmark analysis, examining the impact of SGLT-2i use within 2, 4, and 6 weeks, and 2, 6, and 12 months on mortality, MACE, and MAKE, also demonstrated significant improvements (Table 2). Considering the high number of patients with mortality and ongoing dialysis within the 1-3 months post-transplant, we also attempted to refine our cohort by utilizing different diagnosis or procedure codes for exclusion and then re-calculated our statistics. Despite these adjustments, the results still indicated a significant improvement in primary and secondary outcomes among SGLT-2i users (Supplemental Table S4).

To further assess the comprehensiveness of our study, we conducted long-term risk analyses at 1 year, 3 years, and 5 years post-transplant (Supplemental Table S3). These results demonstrate significant differences in all-cause mortality between SGLT-2i users and non-users across various time points. We also extended our comparative analysis of outcomes between two groups of KTR: those who used SGLT-2i during the first 3 months post-transplant and continued usage from 3 to 6 months, and those who did not use SGLT-2i during the first 3 months and continued non-usage from 3 to 6 months (Supplemental Table S5). Patients were categorized based on the presence of medication codes during these time periods to determine continued usage or non-usage of SGLT-2i. The aHR for all-cause mortality among SGLT-2i users was 0.47 (95% CI: 0.30–0.73, $p < 0.001$) compared to non-users. For MACE, the aHR was 0.66 (95% CI: 0.49–0.90, $p = 0.008$), and for MAKE, the aHR was 0.56 (95% CI: 0.45–0.70, $p < 0.001$).

In the specificity analysis, our study reveals differences between the SGLT-2i users and non-users after PSM. Notably, the frequency of

Table 1 | Baseline characteristics of SGLT-2i users or non-users before and after propensity score matching[a]

| | Before matching | | | After matching | | |
|---|---|---|---|---|---|---|
| | SGLT-2i users (n = 1995) | SGLT-2i non-users (n = 26,834) | Std. Diff. | SGLT-2i users (n = 1970) | SGLT-2i non-users (n = 1970) | Std. Diff. |
| **Demographics** | | | | | | |
| Age at index date | 59.5 ± 11.3 | 57.2 ± 12.3 | 0.20 | 59.5 ± 11.3 | 59.6 ± 11.5 | 0.01 |
| Male, n (%) | 1265 (63.4%) | 15,658 (58.6%) | 0.10 | 1246 (63.2%) | 1246 (63.2%) | <0.01 |
| White, n (%) | 719 (36.0%) | 13,343 (49.9%) | 0.28 | 715 (36.3%) | 690 (35.0%) | 0.03 |
| Not Hispanic or Latino, n (%) | 1298 (65.1%) | 17,597 (65.8%) | 0.02 | 1280 (65.0%) | 1284 (65.2%) | <0.01 |
| **Comorbidities, n (%)** | | | | | | |
| Hypertension | 1720 (86.2%) | 19,980 (74.7%) | 0.29 | 1696 (86.1%) | 1712 (86.9%) | 0.02 |
| Dyslipidemia | 1397 (70.0%) | 14,191 (53.1%) | 0.35 | 1375 (69.8%) | 1399 (71.0%) | 0.03 |
| Obesity | 596 (29.9%) | 5658 (21.2%) | 0.20 | 590 (29.9%) | 597 (30.3%) | 0.01 |
| Heart failure | 417 (20.9%) | 3803 (14.2%) | 0.18 | 412 (20.9%) | 430 (21.8%) | 0.02 |
| Neoplasms | 473 (23.7%) | 5079 (19.0%) | 0.12 | 464 (23.6%) | 444 (22.5%) | 0.02 |
| Liver diseases | 260 (13.0%) | 3145 (11.8%) | 0.04 | 256 (13.0%) | 259 (13.1%) | <0.01 |
| Chronic lower respiratory diseases | 213 (10.7%) | 2709 (10.1%) | 0.02 | 213 (10.8%) | 211 (10.7%) | <0.01 |
| Systemic connective tissue disorders | 66 (3.3%) | 796 (3.0%) | 0.02 | 63 (3.2%) | 65 (3.3%) | 0.01 |
| DM ophthalmology | 291 (14.6%) | 3556 (13.3%) | 0.04 | 289 (14.7%) | 282 (14.3%) | 0.01 |
| DM neuropathy | 387 (19.4%) | 4109 (15.4%) | 0.11 | 384 (19.5%) | 376 (19.1%) | 0.01 |
| DM nephropathy | 1175 (58.9%) | 10,950 (41.0%) | 0.36 | 1,153 (58.5%) | 1164 (59.1%) | 0.01 |
| Nephrotic syndrome | 38 (1.9%) | 359 (1.3%) | 0.04 | 38 (1.9%) | 40 (2.0%) | 0.01 |
| Cystic kidney disease | 117 (5.9%) | 1284 (4.8%) | 0.05 | 116 (5.9%) | 108 (5.5%) | 0.02 |
| Smoking | 18 (0.9%) | 276 (1.0%) | 0.01 | 18 (0.9%) | 17 (0.9%) | 0.01 |
| **Medications, n (%)** | | | | | | |
| Corticosteroids | 1534 (76.9%) | 18,413 (68.9%) | 0.18 | 1,513 (76.8%) | 1518 (77.1%) | 0.01 |
| ACEi/ARB | 878 (44.0%) | 6883 (25.7%) | 0.39 | 860 (43.7%) | 872 (44.3%) | 0.01 |
| Beta-blockers | 1376 (69.0%) | 16,372 (61.2%) | 0.16 | 1,356 (68.8%) | 1387 (70.4%) | 0.03 |
| CCB | 1228 (61.6%) | 13,806 (51.6%) | 0.20 | 1211 (61.5%) | 1211 (61.5%) | <0.01 |
| Diuretics | 1167 (58.5%) | 14,295 (53.5%) | 0.10 | 1154 (58.6%) | 1204 (61.1%) | 0.05 |
| HMG-CoA reductase inhibitors | 1306 (65.5%) | 12,418 (46.4%) | 0.39 | 1284 (65.2%) | 1286 (65.3%) | <0.01 |
| Insulin | 1329 (66.6%) | 15,378 (57.5%) | 0.19 | 1310 (66.5%) | 1362 (69.1%) | 0.06 |
| Biguanides | 468 (23.5%) | 1457 (5.4%) | 0.53 | 450 (22.8%) | 435 (22.1%) | 0.02 |
| Thiazolidinediones | 102 (5.1%) | 303 (1.1%) | 0.23 | 100 (5.1%) | 93 (4.7%) | 0.02 |
| Sulfonylureas | 282 (14.1%) | 2,013 (7.5%) | 0.21 | 276 (14.0%) | 275 (14.0%) | <0.01 |
| GLP-1 agonists | 306 (15.3%) | 763 (2.9%) | 0.44 | 286 (14.5%) | 296 (15.0%) | 0.01 |
| DPP-4 inhibitors | 346 (17.3%) | 1583 (5.9%) | 0.36 | 333 (16.9%) | 331 (16.8%) | <0.01 |
| **Laboratory** | | | | | | |
| Sodium, mmol/L | 139 ± 3.12 | 139 ± 3.36 | 0.01 | 139 ± 3.11 | 139 ± 3.43 | 0.01 |
| Potassium, mmol/L | 4.42 ± 0.552 | 4.42 ± 0.566 | <0.01 | 4.42 ± 0.551 | 4.39 ± 0.553 | 0.06 |
| eGFR[b], mL/min/1.73m$^2$ | 54 ± 22.5 | 51.4 ± 24.5 | 0.11 | 54 ± 22.5 | 53.5 ± 24.9 | 0.02 |
| HbA1c, % | 7.28 ± 1.56 | 6.83 ± 1.7 | 0.28 | 7.27 ± 1.56 | 7.23 ± 1.76 | 0.03 |
| ALT, U/L | 24.1 ± 23.9 | 25.4 ± 31.5 | 0.05 | 24.2 ± 24 | 24.3 ± 21.3 | 0.01 |

**Table 1 (continued) | Baseline characteristics of SGLT-2i users or non-users before and after propensity score matching[a]**

| | Before matching | | | After matching | | |
|---|---|---|---|---|---|---|
| | SGLT-2i users (n = 1995) | SGLT-2i non-users (n = 26,834) | Std. Diff. | SGLT-2i users (n = 1970) | SGLT-2i non-users (n = 1970) | Std. Diff. |
| AST, U/L | 21.2 ± 16.5 | 22.5 ± 24.7 | 0.06 | 21.2 ± 16.6 | 21.9 ± 15 | 0.04 |
| T-Bilirubin, mg/dL | 0.56 ± 0.33 | 0.57 ± 0.71 | 0.02 | 0.56 ± 0.33 | 0.58 ± 0.62 | 0.05 |
| LDL, mg/dL | 79 ± 34.3 | 82.6 ± 36.8 | 0.10 | 78.9 ± 34.3 | 82.1 ± 37.8 | 0.09 |
| HDL, mg/dL | 44.7 ± 17.1 | 45.9 ± 18.3 | 0.07 | 44.7 ± 17.2 | 45.1 ± 17.3 | 0.02 |
| TG, mg/dL | 172 ± 132 | 162 ± 125 | 0.08 | 172 ± 132 | 163 ± 117 | 0.07 |
| Total cholesterol, mg/dL | 156 ± 44.9 | 160 ± 48.4 | 0.09 | 156 ± 45 | 159 ± 48.3 | 0.07 |
| BNP, pg/mL | 597 ± 1,182 | 721 ± 2,257 | 0.07 | 603 ± 1,189 | 732 ± 1,718 | 0.09 |
| SBP, mmHg | 134 ± 19.2 | 134 ± 21 | 0.01 | 135 ± 19.2 | 135 ± 20.9 | 0.01 |
| DBP, mmHg | 73.8 ± 11.5 | 73.3 ± 12.2 | 0.04 | 73.8 ± 11.5 | 73.3 ± 12.4 | 0.04 |
| BMI, kg/m² | 29.7 ± 5.38 | 29.1 ± 5.87 | 0.12 | 29.7 ± 5.39 | 29.9 ± 5.9 | 0.04 |

[a]Plus–minus values are means ± SD.
[b]The Modification of Diet in Renal Disease (MDRD) formula is used to assess the eGFR at baseline.

ACEi angiotensin-converting enzyme inhibitor, ALT alanine aminotransferase, ARB angiotensin II receptor blocker, AST aspartate aminotransferase, BMI body mass index, BNP brain-type natriuretic peptide, CCB calcium channel blocker, DBP diastolic blood pressure, DM diabetes mellitus, DPP-4 inhibitors dipeptidyl peptidase-4 inhibitors, eGFR estimated glomerular filtration rate, GLP-1 agonists glucagon-like peptide-1 agonists, HbA1c hemoglobin A1c, HDL high-density lipoprotein, HMG-CoA Hydroxymethylglutaryl-CoA, LDL low-density lipoprotein, SBP systolic blood pressure, SD standard deviation, SGLT-2i sodium-glucose cotransporter 2 inhibitor, Std. Diff. standardized difference, T-Bilirubin total bilirubin, TG triglycerides.

incident dialysis cases was significantly lower in the SGLT-2i group, as evidenced by an aHR of 0.45 (95% CI: 0.32–0.63). In terms of acute myocardial infarction (AMI), although SGLT-2i users exhibited a lower incidence, the reduction was not statistically significant, with an aHR of 0.72 (95% CI: 0.48–1.08). Similarly, stroke incidence among SGLT-2i users did not significantly differ, showing an aHR of 0.80 (95% CI: 0.52–1.24) (Fig. 2). The use of SGLT-2i does not increase the risk of acute kidney injury (AKI), with an aHR of 0.95 (95% CI: 0.85–1.07). Regarding genitourinary infections, there was no significant increase in the risk for any genitourinary infection (including urinary tract infections [UTI] and candidiasis) with an aHR of 1.03 (95% CI: 0.83–1.28). Specifically, the aHR for UTI alone was 1.02 (95% CI: 0.82–1.27), and for urogenital candidiasis, it was 1.30 (95% CI: 0.74–2.28), indicating no significant risk increase.

Supplemental Table S6 presents a comparison of hemoglobin A1c (HbA1c) levels between users and non-users of SGLT-2 inhibitors at three post-transplant intervals: 3–6 months, 6–9 months, and 9–12 months. After PSM, although the differences at 3–6 months showed higher HbA1c levels in users compared to non-users ($p = 0.049$), at the later intervals (6–9 and 9–12 months), the differences in HbA1c levels were not statistically significant ($p = 0.262$ and $p = 0.393$, respectively).

### Positive and negative controls providing a baseline for comparisons

As ever reported, individuals using SGLT-2i were found to have an increased risk of experiencing diabetic ketoacidosis[14,15] and osteoporotic fractures[16], with aHRs of 1.84 (95% CI: 1.06–3.20) and 2.93 (95% CI: 1.16–7.43), respectively. However, for other conditions such as appendicitis, hematologic malignancy, common cold, irritable bowel syndrome, burn, and suicide attempt and ideation, there were no significant differences in risk between SGLT-2i users and non-users (Fig. 2). Our positive exposure control included patients treated with angiotensin-converting enzyme inhibitors (ACEi) and angiotensin II receptor blockers (ARB). In this group, we observed a significant reduction in all-cause mortality among SGLT-2i users, with an aHR of 0.70 (95% CI: 0.63–0.78). Similar patterns were noted in the MACE and MAKE, with aHRs of 0.83 (95% CI: 0.75–0.91) and 0.87 (95% CI: 0.82–0.93) respectively. Conversely, in our negative exposure control group, which was exposed to vitamin C, the variations in all-cause mortality, MACE, and MAKE were not significant (Fig. 2).

### Subgroup analysis of all-cause mortality, MACE, and MAKE

Our research indicates that SGLT-2i significantly improved health outcomes across diverse demographic groups, including variations by timing of kidney transplantation, sex, age, and levels of HbA1c and hemoglobin, resulting in a universal reduction in all-cause mortality (Fig. 3). This beneficial effect, independent of SGLT-2i dosage, extends to patients with different co-morbidities such as gout, obesity, and various kidney functions. Long-term SGLT-2i users also show more significant benefits in all-cause mortality and MACE than new SGLT-2i users after kidney transplants. The risk of mortality is significantly lower among SGLT-2i users with hypertension, no proteinuria, and those on steroids and tacrolimus. Moreover, SGLT-2i' effectiveness in reducing MACE is particularly evident in patients managing type 2 diabetes with insulin, without heart failure, and those on steroids and tacrolimus. A consistent association was observed between the use of SGLT-2i and a lower risk of MAKE among patients, more pronounced in those without proteinuria, not receiving sulfonylureas, and in those concurrently using steroids and tacrolimus.

### Discussion

In current real-world practice, it has been observed that only 6.5% of diabetic KTR utilize SGLT-2i post-transplant. The striking reduction in all-cause mortality among SGLT-2i users, with a rate of 6.56 per 1,000

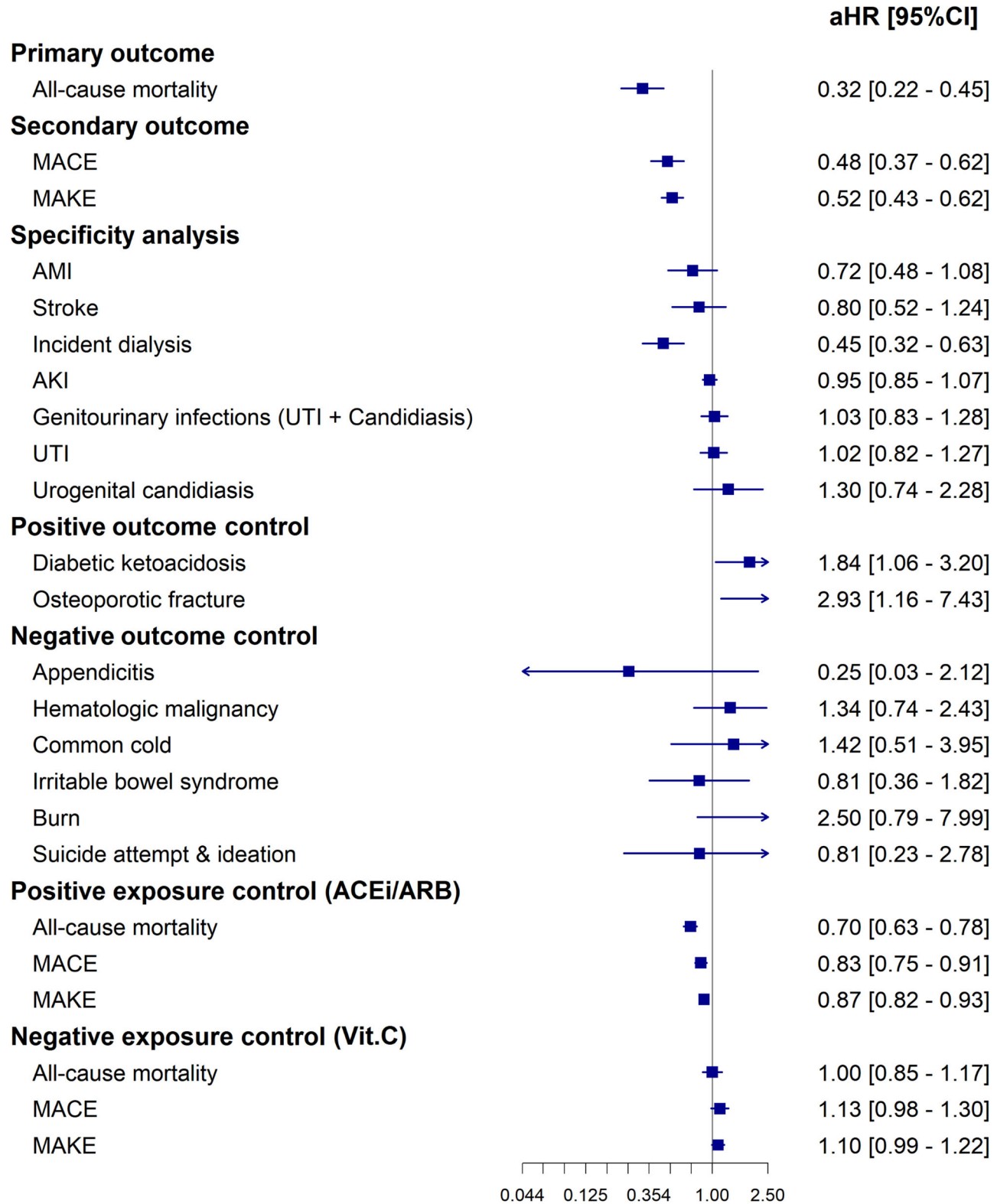

**Fig. 2 | Assessment of primary and secondary outcomes, specificity analysis, and outcome controls in SGLT-2i users (n = 1970) versus non-users (n = 1970) among diabetic KTR.** aHRs and 95% CIs are presented, with the centre defined as the aHR and error bars representing the CIs. The vertical line indicates an aHR of 1.00. ACEi angiotensin-converting enzyme inhibitor; AKI acute kidney injury; aHR adjusted hazard ratio; AMI acute myocardial infarction; ARB angiotensin II receptor blocker; CI confidence interval; KTR kidney transplant recipient; MACE major adverse cardiovascular event; MAKE major adverse kidney event; SGLT-2i sodium-glucose cotransporter 2 inhibitor; UTI urinary tract infection; Vit.C vitamin C.

**Table 2 | Landmark analysis of outcomes of interest at multiple temporal points post-transplant: comparison of SGLT-2i users versus non-users**

| Outcome | aHR (95% CI) | p value[a] |
|---|---|---|
| **Usage of SGLT-2i within the 2 weeks post-transplant** | | |
| All-cause mortality | 0.35 [0.25–0.49] | <0.001 |
| MACE | 0.55 [0.44–0.69] | <0.001 |
| MAKE | 0.65 [0.57–0.75] | <0.001 |
| **Usage of SGLT-2i within the 4 weeks post-transplant** | | |
| All-cause mortality | 0.31 [0.23–0.41] | <0.001 |
| MACE | 0.53 [0.43–0.65] | <0.001 |
| MAKE | 0.56 [0.50–0.64] | <0.001 |
| **Usage of SGLT-2i within the 6 weeks post-transplant** | | |
| All-cause mortality | 0.68 [0.51–0.90] | 0.007 |
| MACE | 0.60 [0.46–0.77] | <0.001 |
| MAKE | 0.67 [0.57–0.79] | <0.001 |
| **Usage of SGLT-2i within the 2 months post-transplant** | | |
| All-cause mortality | 0.47 [0.35–0.63] | <0.001 |
| MACE | 0.58 [0.45–0.75] | <0.001 |
| MAKE | 0.56 [0.47–0.66] | <0.001 |
| **Usage of SGLT-2i within the 6 months post-transplant** | | |
| All-cause mortality | 0.32 [0.22–0.47] | <0.001 |
| MACE | 0.58 [0.45–0.74] | <0.001 |
| MAKE | 0.50 [0.42–0.59] | <0.001 |
| **Usage of SGLT-2i within the 12 months post-transplant** | | |
| All-cause mortality | 0.36 [0.23–0.56] | <0.001 |
| MACE | 0.65 [0.50–0.84] | <0.001 |
| MAKE | 0.53 [0.45–0.64] | <0.001 |

[a]The log-rank test was used to assess differences in outcome probabilities between cohorts, with significance determined at a two-sided p value < 0.05.

aHR adjusted hazard ratio, CI confidence interval, MACE major adverse cardiovascular event, MAKE major adverse kidney event, SGLT-2i sodium-glucose cotransporter 2 inhibitor.

person-years, highlights the substantial potential of using SGLT-2i more often or routinely to improve the survival of diabetic KTR. The observed decrease in MACE and MAKE incidence among SGLT-2i users is particularly noteworthy. The favorable impact of SGLT-2i extends across various stages of baseline kidney function, encompassing patients with or without ACEi/ARB usage, older or younger individuals, those with or without obesity, and individuals with varying HbA1c levels. It suggests that these inhibitors not only have kidney protective effects but also contribute to improved overall kidney health in diabetic KTR. This finding is crucial, given the heightened risk of kidney-related complications in this population. Moreover, our analysis revealed that the beneficial impact of SGLT-2i is consistent across different dosages. This effect ensures that a wide range of patients, including those requiring lower dosages due to impaired kidney function or side effects, can still achieve significant health benefits. The specificity analysis provided further validation of these benefits. The reduced frequency of incident dialysis cases among SGLT-2i users emphasizes the potential of these drugs in preventing the progression to severe kidney complications in diabetic KTR.

The exposure control analyses underscore the reliability of our findings. The reduced mortality and adverse events in patients treated with ACEi and ARB, contrasted with the less pronounced variations in the vitamin C exposed group, add a layer of confidence to our results. Moreover, our application of the E-value indicates that a more substantial unmeasured confounder is unlikely to be necessary to challenge the estimated effect of the covariate, thus strengthening the argument for its causal influence[17]. Landmark analysis, which involved initiating the follow-up period at different timeframes, also showed the

persistent benefit of SGLT-2i, even with intermittent use. This ensures the robustness of our results, mitigating the possibility of guarantee-time bias or immortal time bias[18–20]. Collectively, the findings mentioned above support the integration of SGLT-2i into the treatment regimen for diabetic KTR, given their potential to significantly reduce mortality and adverse cardiac and kidney events. The importance of precision medicine is particularly highlighted in the management of patients with diverse health profiles and comorbidities.

Our study includes significant numbers of patients with pre-transplant diabetes and those diagnosed with PTDM, both major factors affecting morbidity and mortality. According to the 2024 International Consensus on Post-Transplantation Diabetes Mellitus[21], PTDM increases the risk of graft loss, cardiovascular events, and mortality. Similarly, pre-transplant diabetes predicts higher mortality and cardiovascular incidents. Kuo et al.[4] found pre-transplant diabetes to be a primary predictor of all-cause and cardiovascular mortality, highlighting its significance in transplant outcomes. Tsai et al.'s retrospective study[22] of 427 KTR also demonstrated that pre-transplant diabetic patients had a 6.36-fold increased risk of adverse long-term outcomes such as serum creatinine doubling, graft failure, and patient survival, compared to non-diabetic patients. These studies confirm the critical impact of diabetes status on KTR, emphasizing the need for targeted management strategies.

Recent developments in treating type 2 diabetes, especially through SGLT-2i, have shown impressive cardiorenal advantages[5–8]. The significantly lower incidence of MACE in the SGLT-2i group aligns with the growing body of evidence supporting the cardiovascular benefits of SGLT-2i in diabetic patients[10]. The cardiorenal advantages of SGLT-2i stem from multiple interconnected mechanisms, primarily including natriuresis and the subsequent reduction in blood pressure, which substantially enhance cardiac and vascular functions[23–25]. These effects also aid in nephroprotection by increasing tubule-glomerular efficiency. SGLT-2i have been observed to diminish intraglomerular hypertension and hyperfiltration, reduce albuminuria, and augment erythropoietin production, thereby decelerating nephropathy progression[23–25]. Our study supports these outcomes, indicating that SGLT-2i effectively lowered the incidence of MACE and MAKE in KTR. With their extensive benefits, SGLT-2i may be emerging as an essential medication for long-term cardiorenal protection in diabetic KTR. Contrarily, a previous meta-analysis by Wu et al. on SGLT-2i in type 2 diabetes noted protective cardiovascular benefits in high-risk type 2 diabetes patients; however, it observed no significant impact of these inhibitors on the incidence of non-fatal myocardial infarction or unstable angina, while noting an increased risk of non-fatal stroke[26]. A more recent meta-analysis involving 42,568 participants also suggested that although SGLT-2i reduce the overall incidence of MACE, they do not significantly alter the rates of individual myocardial infarction or stroke events[27]. These findings are consistent with our research outcomes, further affirming the significance of SGLT-2i in long-term cardiorenal protection.

Our study showed that while early post-transplant SGLT-2i users exhibited higher initial HbA1c levels than non-users, this disparity diminished over time, at 6-9 and 9-12 months, indicating successful long-term glycemic control. Long-term users particularly benefited from reduced all-cause mortality and MACE compared to newer users, underscoring the pleiotropic effects[28] beyond their initial therapeutic target of glycemic control, while new users still benefit from a reduction in MAKE, which are important events after kidney transplant.

Research also indicates that SGLT-2i plays a crucial role in decelerating the progression of diabetic kidney disease, as evidenced by multiple studies[7–9]. These inhibitors also demonstrate efficacy in managing acute kidney disease (AKD). In a study utilizing the TriNetX database, Pan et al. reported significant reductions in mortality, MACE, and MAKE among AKD patients using SGLT-2i compared to non-users[10]. This underscores the importance of SGLT-2i in mitigating cardiovascular and kidney disease risks in individuals with type 2

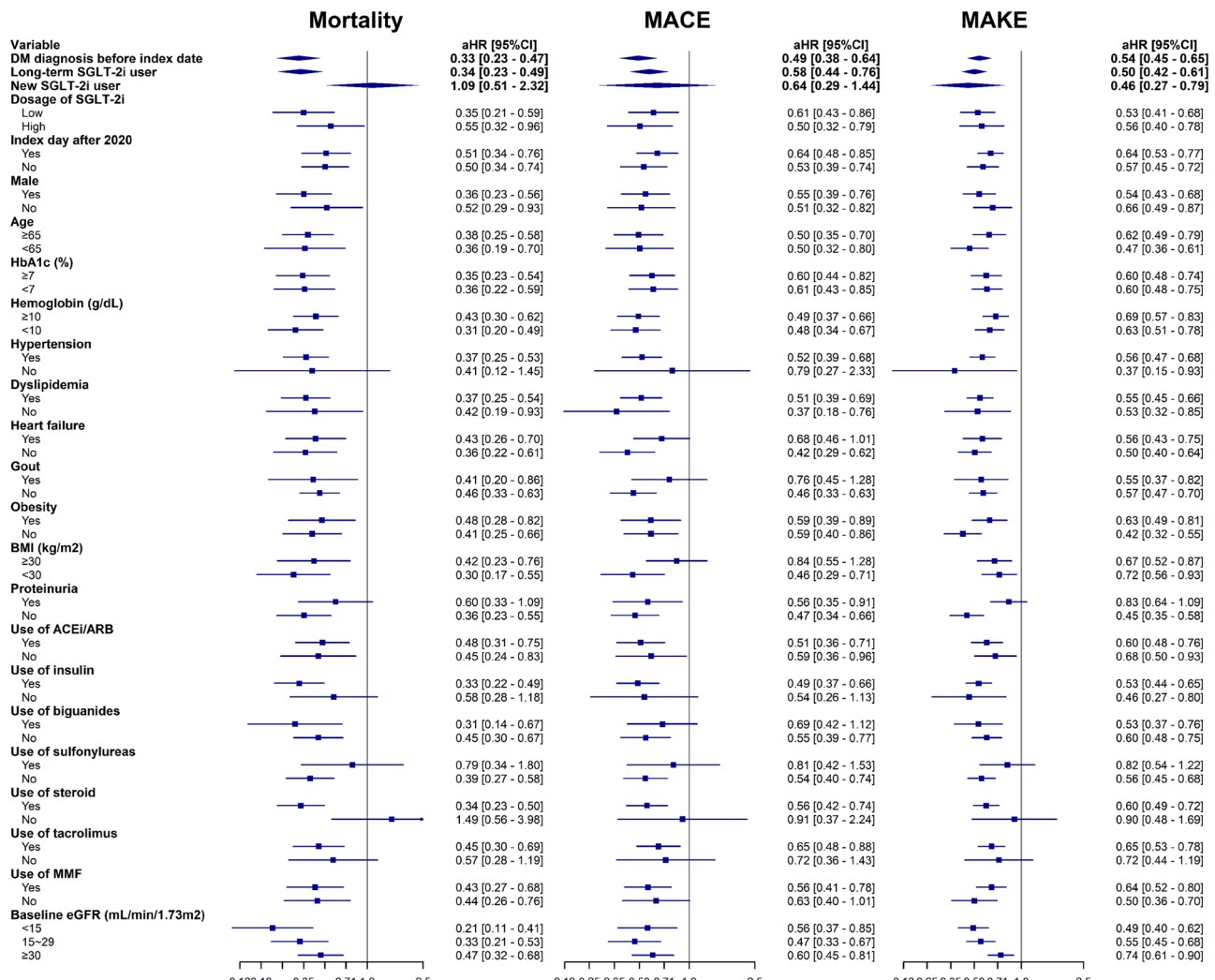

**Fig. 3 | Subgroup analysis.** Forest plots of aHRs for SGLT-2i users (*n* = 1,970) versus non-users (*n* = 1,970) after kidney transplant, regarding the long-term risks assessed in a sensitivity analysis for all-cause mortality, MACE, and MAKE. aHRs and 95% CIs are presented, with the centre defined as the aHR and error bars representing the CIs. The vertical line indicates an aHR of 1.00. Baseline characteristics were assessed based on data collected from one year before and up to the day before the index date; medication usage was evaluated within the initial 90 days following the index date. Definitions for SGLT-2i user types and dosages are as follows: A 'Long-term SGLT-2i user' is defined as a patient who was already using SGLT-2i before the transplant, whereas a 'New SGLT-2i user' refers to a patient who began using SGLT-2i after their transplant. Regarding dosage, 'Low dose SGLT-2i' includes dapagliflozin 5 mg/once daily, canagliflozin 100 mg/once daily, empagliflozin 10 mg/once daily, or ertugliflozin 5 mg/once daily. 'High dose SGLT-2i' is defined as dapagliflozin 10 mg/once daily, canagliflozin 300 mg/once daily, empagliflozin 25 mg/once daily, or ertugliflozin 15 mg/once daily. ACEi angiotensin-converting enzyme inhibitor; aHR adjusted hazard ratio; ARB angiotensin II receptor blocker; BMI body mass index; CI confidence interval; eGFR estimated glomerular filtration rate; HbA1c hemoglobin A1c; MACE major adverse cardiovascular event; MAKE major adverse kidney event; MMF mycophenolate mofetil; SGLT-2i sodium-glucose cotransporter 2 inhibitor.

diabetes and AKD, and may possibly contribute to advantages in KTR, as suggested by the findings in our study.

However, experience with diabetic KTR has been limited, primarily due to concerns over UTI and the risk of hypotension due to osmotic diuresis, which have led to a scarcity of prospective trials in KTR[12,29,30]. Nonetheless, recent studies indicate the efficacy and safety of SGLT-2i in diabetic KTR[31]. Demir et al. observed no increased risk of genital or urinary tract infections with post-transplant SGLT-2i use, alongside a significant reduction in proteinuria and no adverse impact on graft function over a year, in a study conducted across two centers[32]. Similarly, our study also shows that the use of these medications does not increase the risk of genitourinary infections or AKI, confirming their safety profile in this population. A multicenter study by Lim et al. involving diabetic KTR showed improved outcomes related to all-cause mortality, death-censored graft failure, and serum creatinine doubling, with only a few experiencing a temporary estimated glomerular filtration rate-dip[11]. Research by Sánchez Fructuoso et al. demonstrated significant improvements in body weight, blood pressure, blood glucose levels, and kidney function sustained over 6 to 12 months with SGLT-2i treatment[12]. These collective findings suggest that SGLT-2i are not only effective in managing various health aspects in diabetic KTR but also provides a valuable option for long-term care. Their role in enhancing cardiorenal outcomes, together with a manageable safety profile, make SGLT-2i a vital therapeutic choice for diabetic KTR.

The primary strength of this study lies in its comprehensive investigation of the effects of SGLT-2i on diabetic KTR. Utilizing the extensive patient data from the TriNetX electronic health record database, the study boasts a robust and diverse sample size, enabling a thorough and nuanced analysis of the implications of SGLT-2i on this particular patient group. Additionally, the use of the PSM and *E*-value in our research methodology enhances the reliability and validity of

our findings, providing a more robust basis for our conclusions. However, it's critical to acknowledge certain limitations might affect the interpretation of our results. First, the reliance on diagnostic codes for identifying conditions introduces a risk of ascertainment bias, potentially leading to an under-representation of certain conditions. Second, the retrospective nature of this study, dependent on diagnostic and treatment codes, is subject to mis-classification bias and residual confounding, which cannot be entirely eliminated despite the implementation of rigorous methodologies and variable validation strategies. Third, the exclusion of cases with incomplete outcome data might induce selection bias. The high number and percentage of excluded cases were primarily due to our stringent criteria for ensuring data integrity. To avoid potential biases and inaccuracies in morbidity classifications, we excluded all patients with any dialysis-related diagnostic or procedure codes within 1 to 3 months post-transplant, potentially contributing to initially high observed rates of mortality and morbidity post-transplant in our study. Fourth, the absence of specific dates for SGLT-2i usage within the initial 90 days post-transplant limits our ability to fully evaluate early use implications. Additionally, while accounting for biases like selective prescription for recurrent UTI patients, direct evidence on decision-making for SGLT-2i use remains unavailable. Fifth, immortal time bias represents a limitation in our study, as discrepancies between the length of follow-up and duration of SGLT-2i use can skew results. To address this, we conducted landmark analyses across various selection periods, ensuring consistent and robust findings. Lastly, it is important to note that our data source is aggregate data rather than individual-level data, which restricts us from employing more delicate designs, such as randomized controlled trial emulation, prescription time-distribution matching, and time-varying exposure models (like the clone-censor-weight approach)[33]. We are generally concerned about the use of aggregated data for conducting comparative effectiveness research; however, we acknowledge that no other databases offer this level of detail, and some evidence is better than none. To compensate for this limitation, we conducted extensive sensitivity analyses, including long-term risk assessments at multiple time points, to help ensure the robustness and reliability of our findings despite the constraints of using aggregate data.

In conclusion, our observations implicate that SGLT-2i usage among diabetic KTR is associated with a lower risk of all-cause mortality, MACE, and MAKE, as well as a reduced incidence of dialysis, indicating a helpful renoprotective role. These findings highlight the potential of SGLT-2i in enhancing post-transplant care and suggest a transformative impact on treatment strategies. However, while our results are highly implying the benefits of these drugs, it is imperative to conduct a formal randomized controlled trial to confirm these findings definitively. Future prospective studies should further explore the long-term effects and safety profile of SGLT-2i in diabetic KTR, thereby expanding our understanding of their impact on various at-risk populations. Given the significant potential of SGLT-2i, we advocate for their judicious clinical use, coupled with personalized approaches to optimize outcomes in diabetic KTR.

## Methods

This study complied with all relevant ethical regulations and was approved by the Institutional Review Board (IRB) of Chi-Mei Medical Center, Tainan, Taiwan (Approval Numbers: 11202-002, 11210-E01). Written informed consent was waived by the IRB due to the retrospective nature of the study. The study adheres to the ethical principles delineated in the Declaration of Helsinki[34], emphasizing the utmost importance of protecting the rights and welfare of research subjects.

This investigation utilized the TriNetX database, a platform dedicated to upholding the standards established by the Health Insurance Portability and Accountability Act and the General Data Protection Regulation, ensuring maximum security for patient data[10]. The Western Institutional Review Board granted a waiver for informed consent, justified by TriNetX's ability to generate only aggregated and statistical summaries from de-identified data acquired from various healthcare providers. This method effectively safeguards patient privacy and confidentiality.

### Study design and data source

This research utilized a historical cohort dataset obtained from the TriNetX network, a significant collection predominantly made up of electronic health record data, enriched with additional laboratory and mortality information. The dataset extends up to the present and includes data on over 100 million individuals. It provides a diverse range of data, including patient demographics, medical diagnoses (coded using ICD-10-CM), various medical procedures (recorded with ICD-10 Procedure Coding System, Current Procedural Terminology codes, or Systematized Nomenclature of Medicine - Clinical Terms), prescribed drugs (categorized by the Anatomical Therapeutic Chemical Classification system or RxNorm), laboratory tests (classified by Logical Observation Identifiers Names and Codes), genetic data (notated according to the Human Genome Variation Society standards), and detailed records of healthcare service use. This dataset was securely accessed for this study in January 2024 and includes data collected from June 1, 2015, to June 1, 2023, within the TriNetX Network.

### Study population and cohort construction

In our research, we focused on KTR diagnosed with type 2 diabetes, as classified under the ICD-10-CM code E11. The index date for these patients was defined as the date of their kidney transplant, determined using the ICD-10-CM code Z94.0, along with associated procedure codes for kidney transplantation. Our inclusion criteria encompassed patients with type 2 diabetes, aged over 18 years, who were administered SGLT-2i within the 3 months post-transplant. The study's control group comprised diabetic KTR who did not receive SGLT-2i within the 3 months post-transplant. We deliberately excluded patients in both the user and non-user groups who died or required dialysis between 1 to 3 months post-transplant, as dialysis dependency during this period may indicate graft failure, alongside the restriction on the use of SGLT-2i in patients undergoing dialysis[35,36]. This exclusion aligns with the landmark method[18], minimizing the possibility of an immortal time interval[20].

### Outcome measures and covariates

This study primarily aimed to evaluate all-cause mortality, while also closely monitoring MACE, and MAKE as secondary outcomes. The MACE we observed included cerebral infarction, hemorrhagic stroke, AMI, cardiac arrest, and death. MAKE, on the other hand, encompassed events such as the initiation of re-dialysis, incident dialysis, or death. These outcomes were meticulously tracked from the 90th day post the index date and up to 5 years. To mitigate protopathic or ascertainment bias, any events of secondary outcomes except MAKE that occurred before the index date were excluded, and repeat PSM was performed.

In addition to our primary and secondary outcome analyses, our study conducted detailed specificity analyses to further investigate the components of MACE and MAKE. We also evaluated potential side effects associated with SGLT-2i, ensuring a comprehensive understanding of the therapy's safety profile. To bolster the credibility of our findings, we implemented both positive and negative outcome controls. For a balanced comparison between the study groups, we incorporated a variety of demographic factors such as age, sex, and ethnic and racial backgrounds. Moreover, we scrutinized various comorbidities and concurrent medication use, identified through ICD-10-CM codes, the Anatomical Therapeutic Chemical Classification system, or RxNorm (Supplementary Appendix).

## Prespecified analyses of subgroups

To explore potential variations in risk associated with desired outcomes among users of SGLT-2i, prespecified subgroup analyses were conducted. These analyses considered several predetermined factors, such as age, hypertension, heart failure, pre-transplant estimated glomerular filtration rate, proteinuria, the period of enrollment, previous diagnosis of type 2 diabetes before the index date, concurrent use of other medications for glycemic control, and the use of renin-angiotensin-aldosterone system blockers and immunosuppressive therapy.

## Statistical analysis

To achieve an equitable distribution of covariates across the primary study group and a comparison group, our approach involved using TriNetX's built-in statistics to match the two groups in a 1:1 PSM[37]. This method was applied to a comprehensive set of 45 characteristics, covering a wide range of categories, including patient demographics (such as age and sex), medical history (e.g., previous diagnoses), concomitant medications, and laboratory test results. This process was conducted in two separate cohorts. The matching utilized a greedy nearest-neighbor algorithm, with a caliper limit set at 0.1 of the pooled standardized difference[38]. This approach ensured a similar distribution of covariates across both groups.

In our data analysis, we categorized variables as either counts and percentages, or as means with their corresponding standard deviations. We estimated survival probabilities using the Kaplan-Meier method. We conducted multiple sensitivity and specificity analyses to validate the reliability of our results. aHR and 95% CI were derived using Cox proportional hazard regression models[39]. The hazard ratios were adjusted for age, sex, and race due to their potential interactions with kidney disease. The log-rank test was utilized to identify any significant differences in outcome probabilities between our cohorts. Significance was determined at a two-sided *p*-value of less than 0.05. Furthermore, to address potential unmeasured confounders, we calculated *E*-values for both primary and secondary outcomes[40]. We also performed a sequential landmark analysis by setting multiple landmark times to verify the stability of the results[19]. Analytical tools employed included R software (version 3.2.2, Free Software Foundation, Boston, MA).

## Reporting summary

Further information on research design is available in the Nature Portfolio Reporting Summary linked to this article.

# Data availability

The aggregated datasets generated and analyzed during this study were obtained from the TriNetX platform. Due to TriNetX's data-sharing policies, we do not have access to individual-level data; only de-identified, aggregated data were available to us for analysis. Access to the TriNetX data is restricted as it contains protected health information. Researchers wishing to access these data may apply through the TriNetX platform. Approval for access typically requires demonstration of appropriate credentials, a clear research purpose, and compliance with applicable privacy regulations. The application and approval process may require several weeks to complete, subject to the applicant's qualifications and the specifics of the research proposal. For more information on data access, please visit the TriNetX website: https://trinetx.com, or contact TriNetX directly via their support email at support@trinetx.com. Source data are provided with this paper.

# Code availability

Basic statistical analyses in this study were conducted using the built-in statistical system of the TriNetX platform. The additional code supporting this study is publicly available on GitHub at https://github.com/Fangyu618/my-R-project.git.

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

## Acknowledgements

The authors greatly appreciate the Second Core Lab in National Taiwan University Hospital for technical assistance. We also express our sincere gratitude to all staff of the Taiwan Clinical Trial Consortium (TCTC). This study was supported by Ministry of Science and Technology (MOST) of the Republic of China (Taiwan) [grant number, MOST 107-2314-B-002-026-MY3, 109-2314-B-002-174-MY3, 110-2314-B-002-124-MY3, 110-2314-B-002-241, 110-2314-B-002-239], National Science and Technology Council (NSTC) [grant number, NSTC 111-2314-B-002-046, 111-2314-B-002-058, 111-2314-B-002-232-MY3, 112-2314-B-002-029, 112-2314-B-002-040], National Taiwan University Hospital [109-S4634, PC-1246, PC-1309, PC-1446, VN109-09, UN109-041, UN110-030, 111-FTN0011, 112-FTN0010, 112-S0088, 112-UN0018, 113-S0173, 113-L1004], Grant MOHW 110-TDU-B-212-124005, and Mrs. Hsiu-Chin Lee Kidney Research Fund. The funding and assistance were awarded to the author V.C.W.

## Author contributions

The conception and design of the study were undertaken by J.Y.S. and V.C.W. The acquisition of data was performed by J.Y.S. and L.Y.C., while the analysis and interpretation of the data were conducted by J.Y.S. The initial drafting of the manuscript was executed by J.Y.S. and V.C.W. Critical revision of the manuscript for important intellectual content was undertaken by H.C.P. and J.S.C. The statistical analysis was conducted by J.Y.S. and V.C.W. Funding acquisition was managed by V.C.W. Administrative, technical, or material support was provided by J.Y.C. Supervision was provided by C.S.T. and J.S.C. All authors contributed to the manuscript review process and approved the final version for submission.

## Competing interests

The authors declare no competing interests.
