## [Transparent Peer Review file · Nature Communications]

The Outcomes of SGLT-2 Inhibitor Utilization in Diabetic Kidney Transplant Recipients

Corresponding Author: Professor Jeff Chueh

Version 0:

Reviewer comments:

Reviewer #1

(Remarks to the Author)

This is a study trying to answer an important question. Do SGLT-2i drugs offer kidney transplant recipients benefit against cardiovascular or renal morbidity and mortality, similar to that of their effect in people with Type 2 diabetes as demonstrated by a number of large scale clinical trials. The results are important to the kidney transplant community.

I need some clarity over the following:

1. Please make clearer the population studied. Did they have pre- and/or post transplant diabetes? Did all patients with Type 2 diabetes get included? PTDM is officially not Type 2 diabetes, but a diagnosis of its own. Please make clear whether you included people with pre-transplant diabetes, or only PTDM.
2. A reason not to prescribe SGLT-2i is in patients with recurrent UTI. Could some patients have deliberately not been given these drugs due to uro infection issues?
3. Your initial cohort had 3249 patients on SGLT-2i, of who only 1995 were included in the analysis. Did 1254 (38.5%) patients really die or go on to dialysis within three months? This seems a very high rate or mortality / morbidity from kidney transplantation?
4. The results do look remarkable with regards to mortality, MACE and MAKE. Could there be any bias in prescribing of SGLT-2i to people who were fitter or had fewer comorbidities? What about prescribing of other CV drugs such as ACEI/ARB, statins. What about glucose control?
5. Was there any increase in genitourinary infections (UTI, candidiasis)?
6. The authors must in their conclusions state that whilst their results are highly suggestive of a positive benefit of these drugs in KTR, it is imperative that a formal randomised controlled trial is done to confirm the findings.

Reviewer #2

(Remarks to the Author)

JY Sheu and colleagues have conducted a necessary study on the long-term effects of SGLT-2 inhibitors, which are known for their glucose-regulating, cardioprotective, and kidney protective effects in a retrospective large cohort of transplant patients. However, the authors need to address the following concerns:

Major Criticisms:

1. It is unclear if the study specifically focuses on post-transplant diabetes mellitus (PTDM) as the subjects described are diagnosed with type 2 diabetes. Clarification is needed on whether patients who already had diabetes before their transplant were included. If patients with pre-existing diabetes constitute a significant portion of the study, the title "The Outcomes of SGLT-2 Inhibitor Utilization in Kidney Transplant Recipients with Post-Transplant Diabetes Mellitus KTR diagnosed with type 2 diabetes" needs revision.

2. Additionally, aligning the index date of the PTDM study with the date of kidney transplantation is incorrect. The index date should be adjusted to the date when PTDM was diagnosed.
3. If the study correctly addresses PTDM, the authors' assertion that more than 50% of the patients had diabetic nephropathy at the baseline index date (Table 1) is peculiar and needs clarification.
4. The authors should demonstrate the differential effects of SGLT-2 inhibitor use between patients with diabetes before transplantation and those who developed diabetes post-transplantation.
5. The actual extent of SGLT-2 inhibitor usage, how long after transplantation they were used, and the possibility of the effects being attributable to other medications like metformin need to be explored. Typically, SGLT-2 inhibitors are used in patients with good renal function, so it's crucial to assess if the patients had inherently good prognoses at the time of drug initiation or PTDM diagnosis, not just at transplantation.
6. There is a significant concern about immortal time bias—some patients might have longer follow-up but shorter duration of SGLT-2 inhibitor use, while others might have shorter follow-up but longer usage duration. It's also possible that some patients did not use SGLT-2 inhibitors at the start of follow-up. The study needs to clarify how it dealt with this issue, or otherwise, sufficient statistical adjustments are required.

Minor Criticisms:

1. It is important to assess the risks of urinary tract infections, euglycemic diabetic ketoacidosis, and acute kidney injury associated with SGLT-2 inhibitor use, if not already done.
2. The term "renal" should be replaced with "kidney" throughout the document, i.e., "renal transplantation" should be "kidney transplantation", and "renal replacement therapy" should be "kidney replacement therapy".

Version 1:

Reviewer comments:

Reviewer #1

(Remarks to the Author)

I think the authors have thoughtfully taken on board reviewer comments and dealt with all queries and comments in a thorough manner. I think the manuscript is definitely much improved and ready for publication.

Reviewer #2

(Remarks to the Author)

Thank you for addressing my previous comments so thoroughly. I am satisfied with the comprehensive responses provided, and I appreciate the meticulous attention given to my feedback.

I particularly commend the authors for incorporating landmark analyses, which is an excellent idea to enhance the robustness of the findings. As a final recommendation, I suggest that the authors update the main table to include the landmark analyses results. Additionally, it would be beneficial to describe the landmark analysis methodology in the Statistical Analysis section of the Methods part of the manuscript.

Thank you for your diligent work on these revisions.

Reviewer #3

(Remarks to the Author)

Review for Nature Communications

This study details a retrospective cohort study conducted in the TriNetX Database. Specifically, they compare individuals who initiate an SGLT2-i within the first 3 months post-kidney transplant with individuals who do not. They further exclude individuals who die or have evidence of ongoing dialysis in this 3 month period. The authors have calculated a highly protective hazard ratio of 0.32 for all cause mortality, which is very impressive if true, and would make a big difference to clinical practice. However, I am concerned that their analysis strategy is flawed, and have the following suggestions to address some "self-inflicted" biases (<https://doi.org/10.1016/j.jclinepi.2016.04.014>)

1. The authors should employ a new-user design to avoid selection bias (including individuals who were on SGLT2-is pre-transplant to those who initiated post-transplant).
2. The authors should treat the exposure (initiation of an SGLT2) as a time-varying exposure. This will allow them to include individuals who die or use dialysis in that 3-month period. This can be achieved using the clone-censor-weight approach (<https://doi.org/10.1136/bmj.k182>). This will also allow the authors to use all the data available to them, instead of discarding

the majority of the individuals who do not initiate an SGLT2-i as they have in their propensity-score matched analysis.
3. Hazard ratios are flawed measures (10.1097/EDE.0b013e3181c1ea43). It would be prudent of the authors to present absolute risks (and risk differences), to let readers decide for themselves if there is a clinically meaningful effect. There may be in this scenario, as kidney transplant recipients have a high burden of comorbidities. However, when publishing results that suggest such a big reduction in mortality, the authors would be wise to be more thorough in their analysis strategies.

Together, these three comments may make a big impact on the analysis presented here. The authors have done quite a lot of work in an impressive data source. It would be excellent to see this work in print, however the results as they are written now are probably too good to be true.

The following suggestions may also be prudent to incorporate:

4. While including a negative exposure (Vitamin D) as a control is a good idea, the one selected by the authors is unlikely to have a similar confounding structure to that of the original treatment strategy (SGLT2-i). Individuals choose to initiate Vitamin D for many different reasons than the reasons they get prescribed SGLT2-is.

5. The matching (or outcome adjustment, if the authors choose to alter their analysis strategy) would benefit from including pre-transplant eGFR and pre-transplant blood pressure (SBP and DBP, or MAP), both of which have an effect on treatment choice and mortality. These may have a large association with SGLT2-i initiation and mortality, violating the e-value study the authors have included in Supplemental information.

Reviewer #4

(Remarks to the Author)

Version 2:

Reviewer comments:

Reviewer #3

(Remarks to the Author)

Thank you for your very thoughtful response to our concerns. Only two small points remain:

1. Please add the new user analysis to the supplemental material. Even if it is not sufficiently powered, it is still interesting to see if the point estimates are in the same direction, and whether it is of a similar magnitude, to the point estimates in the main analysis.

2. We are generally concerned about the use of aggregated data to conduct comparative effectiveness research. That said, we recognize that there are not other databases available with this level of detail, and some evidence is better than no evidence. Please add this to the limitations section of the manuscript.

Reviewer #4

(Remarks to the Author)

Reviewer #1:

1. Please make clearer the population studied. Did they have pre- and/or post transplant diabetes? Did all patients with Type 2 diabetes get included? PTDM is officially not Type 2 diabetes, but a diagnosis of it's own. Please make clear whether you included people with pre-transplant diabetes, or only PTDM.

Response: Thank you for your comment requesting further clarification about the population studied. We appreciate your attention to the distinction between PTDM and Type 2 diabetes. In our study, we included all patients with a type 2 diabetes diagnosis, whether pre-existing or that developed post-transplant. Among the 1,995 SGLT-2i users in our study, 66 (3%) were diagnosed PTDM, compared to 2,218 (8%) among the 26,834 SGLT-2i non-users (Line 149-150). Given this inclusion criterion, our cohort actually represents all diabetic kidney transplant recipients (**Diabetic KTR**)^{1, 2} rather than just those exclusively with PTDM. Importantly, our subgroup analysis revealed that in patients with a DM diagnosis before the index date only, the use of SGLT-2i could significantly reduce all-cause mortality, MACE, and MAKE, as demonstrated in our main analysis (**Figure 3**). Reflecting on your feedback, we have revised the terminology from PTDM to Diabetic KTR throughout our manuscript.

Reference:

1. Lim JH, *et al.* The Efficacy and Safety of SGLT2 Inhibitor in Diabetic Kidney Transplant Recipients. *Transplantation* **106**, e404-e412 (2022).
2. Sánchez Fructuoso AI, *et al.* Sodium-glucose cotransporter-2 inhibitor therapy in kidney transplant patients with type 2 or post-transplant diabetes: an observational multicentre study. *Clinical kidney journal* **16**, 1022-1034 (2023).

2. A reason not to prescribe SGLT-2i is in patients with recurrent UTI. Could some patients have deliberately not been given these drugs due to uro infection issues?

Response: We appreciate your attention to the possibility of selective prescription practices

for SGLT-2 inhibitors due to recurrent UTIs. As our study is retrospective and utilizes diagnostic and treatment codes, direct evidence confirming whether decisions to withhold SGLT-2 inhibitors were specifically due to recurrent UTIs is not available. We recognize this as a limitation and will discuss it further in the manuscript (Line 350-352).

Additionally, we have performed extended statistical analyses on the risks of genitourinary infections. Our findings indicate no significant differences in the incidence of genitourinary infections, UTIs, or urogenital candidiasis three months after the index date. The adjusted hazard ratios (aHR) with 95% confidence intervals (CI) are: for any genitourinary infection, aHR 1.03 (95% CI 0.83-1.28); for UTI alone, aHR 1.02 (95% CI 0.82-1.27); and for urogenital candidiasis, aHR 1.30 (95% CI 0.74-2.28). These data have been added to **Figure 2** and elaborated upon in the Results and Discussion sections of our manuscript (Line 187-192, 318-320).

3. Your initial cohort had 3249 patients on SGLT-2i, of who only 1995 were included in the analysis. Did 1254 (38.5%) patients really die or go on to dialysis within three months? This seems a very high rate of mortality / morbidity from kidney transplantation?

Response: Thank you for your concerns regarding the apparently high attrition rate in our cohort within the first three months post-transplantation. The morbidity data in our study, sourced from the TriNetX database, relies on diagnostic and procedure codes. To avoid potential selective biases and ensure the integrity of our analysis, we excluded any patient who had any of the listed dialysis-related codes, even those who could wean from dialysis within 3 months after the index date. This approach was taken to mitigate concerns regarding the accuracy of morbidity classifications and contributed to the seemingly high rates of mortality and morbidity observed in our initial results (Line 342-348).

To address these concerns, we conducted additional analyses focusing exclusively on patients who received both dialysis-related diagnosis and procedure codes within the first

1-3 months post-transplantation. The findings from these additional analyses have significantly changed the patient numbers those were excluded from analysis, but reaffirmed the significant improvements observed in the primary and secondary outcomes, including all-cause mortality, major adverse cardiac events (MACE), and major adverse kidney events (MAKE). We appreciate your diligence in scrutinizing our results, and believe that these additional analyses provide further clarity and assurance regarding the robustness of our findings (**Supplementary Table 4**).

Our study includes a significant proportion of patients with either pre-transplant diabetes or those diagnosed with post-transplant diabetes mellitus (PTDM), both of which are known to significantly influence morbidity and mortality rates. According to the 2024 International Consensus on Post-Transplantation Diabetes Mellitus¹, PTDM is associated with increased risks such as overall graft loss, elevated cardiovascular events, and higher all-cause mortality^{2, 3, 4}. Importantly, pre-transplant diabetes is also a significant predictor of mortality and cardiovascular events in this patient population⁵. A previous study revealed an increased risk of death-censored graft failure by 13.95-fold in the pre-transplant DM Group (95% CI: 2.96-65.75; $p < 0.001$) compared to the non-DM group⁶. Given these significant health risks, the high initial mortality and morbidity rates within our cohort reflect the severe complications associated with diabetes in kidney transplant recipients. We have incorporated your valuable advice and concerns into the Discussion section of our manuscript (Line 264-274).

Reference:

1. Sharif A, *et al.* International consensus on post-transplantation diabetes mellitus. *Nephrology Dialysis Transplantation* **39**, 531-549 (2024).
2. Eide IA, *et al.* Mortality risk in post-transplantation diabetes mellitus based on glucose and HbA1c diagnostic criteria. *Transplant international : official journal of the European Society for Organ Transplantation* **29**, 568-578 (2016).

3. Eide IA, Halden TAS, Hartmann A, Dahle DO, Åsberg A, Jenssen T. Associations Between Posttransplantation Diabetes Mellitus and Renal Graft Survival. *Transplantation* **101**, 1282-1289 (2017).
4. Topitz D, Schwaiger E, Frommlet F, Werzowa J, Hecking M. Cardiovascular events associate with diabetes status rather than with early basal insulin treatment for the prevention of post-transplantation diabetes mellitus. *Nephrology, dialysis, transplantation : official publication of the European Dialysis and Transplant Association - European Renal Association* **35**, 544-546 (2020).
5. Kuo HT, Sampaio MS, Vincenti F, Bunnapradist S. Associations of pretransplant diabetes mellitus, new-onset diabetes after transplant, and acute rejection with transplant outcomes: an analysis of the Organ Procurement and Transplant Network/United Network for Organ Sharing (OPTN/UNOS) database. *American journal of kidney diseases : the official journal of the National Kidney Foundation* **56**, 1127-1139 (2010).
6. Tsai JP, Lian JD, Wu SW, Hung TW, Tsai HC, Chang HR. Long-term impact of pretransplant and posttransplant diabetes mellitus on kidney transplant outcomes. *World J Surg* **35**, 2818-2825 (2011).

4. The results do look remarkable with regards to mortality, MACE and MAKE. Could there be any bias in prescribing of SGLT-2i to people who were fitter or had fewer comorbidities? What about prescribing of other CV drugs such as ACEI/ARB, statins. What about glucose control?

Response: We are grateful for your comment on potential biases in prescribing SGLT-2i. The presence of comorbidities, variation in medication use, and differences in glucose control could significantly impact outcomes such as mortality, MACE, and MAKE. To address these concerns, we employed strict Propensity Score Matching (PSM), including comorbidities, cardiovascular medications (such as ACEI/ARB and statins), and HbA1c levels. After PSM, the standard deviations of baseline characteristics between SGLT-2i users and non-users were all less than 0.1, indicating minimal bias from these factors (**Table 1**). Specifically, for ACEI/ARB usage, the pre-PSM proportions were 878 (44.0%) of SGLT-2i users (n = 1,995) compared to 6,883 (25.7%) of SGLT-2i non-users (n = 26,834), with a standard difference of 0.39. Post-PSM, the proportions were 860 (43.7%) of SGLT-2i users (n = 1,970) compared to 872 (44.3%) of SGLT-2i non-users (n = 1,970), with a standard difference of 0.01. Similarly,

for HMG-CoA reductase inhibitors (statins), the pre-PSM proportions were 1,306 (65.5%) of SGLT-2i users (n = 1,995) compared to 12,418 (46.4%) of SGLT-2i non-users (n = 26,834), with a standard difference of 0.39. Post-PSM, the proportions were 1,284 (65.2%) of SGLT-2i users (n = 1,970) compared to 1,286 (65.3%) of SGLT-2i non-users (n = 1,970), with a standard difference of <0.01. These results indicate that after matching, the groups are well-balanced regarding cardiovascular medications. This rigorous matching process is also evident in our subgroup analysis concerning other oral hypoglycemic agents (OHA), including biguanides, insulin, and sulfonylurea, as depicted in **Figure 3**.

Additionally, we applied the *E*-value analysis to assess the robustness of our findings against potential unmeasured confounding factors (**Supplemental Table S1**). The results suggest that an extremely large unmeasured confounder would be required to nullify the observed associations, further substantiating the reliability of our causal inferences. We also further assessed the effectiveness of glycemic control by comparing HbA1C levels between the groups treated with and without SGLT2 inhibitors at three follow-up intervals: 3-6 months, 6-9 months, and 9-12 months. After PSM, although the differences at 3-6 months showed higher HbA1c levels in users compared to non-users ($p = 0.049$), at the later intervals (6-9 and 9-12 months), the differences in HbA1c levels were not statistically significant ($p = 0.26$ and $p = 0.39$, respectively). We have included these findings in additional **Supplemental Table S5** to provide a comprehensive view of the effects of SGLT2 inhibitors on glycemic control over time (Line 193-198). This underscores the comprehensive cardiorenal effects of SGLT-2 inhibitors beyond their initial therapeutic target of glycemic control. Importantly, these pleiotropic effects include reducing cardiovascular death and heart failure hospitalizations, in addition to potential contributions from intensive glucose control¹ (Line 298-303).

Supplemental Table S5. Comparison of HbA1c Levels Between SGLT-2i Users and Non-Users at 3-6, 6-9, and 9-12 Months

Post-Transplant

	Before matching			After matching		
	SGLT-2i users	SGLT-2i non-users	p -value	SGLT-2i users	SGLT-2i non-users	p -value
Post-transplant HbA1c (%)*						
3 to 6 months	7.5 ± 1.6	7.1 ± 1.7	<0.01	7.5 ± 1.6	7.4 ± 1.6	0.049
6 to 9 months	7.5 ± 1.5	7.1 ± 1.7	<0.01	7.5 ± 1.5	7.4 ± 1.7	0.26
9 to 12 months	7.5 ± 1.5	7.2 ± 1.7	<0.01	7.5 ± 1.5	7.4 ± 1.7	0.39

* Plus-minus values are means ±SD.

Abbreviation:

HbA1c, hemoglobin A1c; SGLT-2i, sodium-glucose cotransporter 2 inhibitor.

Reference:

1. Tanna MS, Goldberg LR. The pleiotropic cardiovascular effects of sodium-glucose cotransporter-2 inhibitors. *Current opinion in cardiology* 36, 764-768 (2021).

5. Was there any increase in genitourinary infections (UTI, candidiasis)?

Response: Thank you for prompting further investigation into genitourinary infections associated with SGLT-2 inhibitor use. Following your suggestion, we conducted detailed statistical analyses to assess the risks of UTI and candidiasis. Our analysis revealed that there were no statistically significant differences in the prevalence of genitourinary infections across the study cohort. Specifically, the adjusted hazard ratios (aHR) with 95% confidence intervals (CI) were: for all genitourinary infections combined, aHR 1.03 (95% CI 0.83-1.28); for UTI alone, aHR 1.02 (95% CI 0.82-1.27); and for urogenital candidiasis, aHR 1.30 (95% CI 0.74-2.28). These outcomes are now included in **Figure 2** and discussed extensively in both the Results and Discussion sections of our manuscript (Line 187-192, 318-320). Your insights have significantly contributed to enhancing the thoroughness of our

analysis.

6. The authors must in their conclusions state that whilst their results are highly suggestive of a positive benefit of these drugs in KTR, it is imperative that a formal randomised controlled trial is done to confirm the findings.

Response: We appreciate your suggestion regarding the need for a formal randomized controlled trial. We have incorporated this recommendation into our Conclusion section. We are grateful for your guidance which underscores the importance of further validating our findings through rigorous research. (Line 361-365)

Reviewer #2:

Major Criticisms:

1. It is unclear if the study specifically focuses on post-transplant diabetes mellitus (PTDM) as the subjects described are diagnosed with type 2 diabetes. Clarification is needed on whether patients who already had diabetes before their transplant were included. If patients with pre-existing diabetes constitute a significant portion of the study, the title "The Outcomes of SGLT-2 Inhibitor Utilization in Kidney Transplant Recipients with Post-Transplant Diabetes Mellitus KTR diagnosed with type 2 diabetes" needs revision.

Response: Thank you for seeking clarification on the focus of our study regarding the inclusion of post-transplant diabetes mellitus (PTDM) and patients diagnosed with type 2 diabetes. We acknowledge the confusion caused by our initial description and appreciate the opportunity to clarify. Our study includes both diabetic patient groups diagnosed with type 2 diabetes before their kidney transplant and those who developed diabetes subsequent to their transplant, commonly referred to as PTDM. Since a significant portion of our cohort consists of patients with pre-existing type 2 diabetes (among the 1,995 SGLT-

2i users in our study, 66 were diagnosed post-transplant, compared to 2,218 among the 26,834 SGLT-2i non-users), we recognize that the term PTDM could not accurately describe our entire study population. In light of this, we decided our study title and description should be revised to more accurately reflect the inclusion of all diabetic kidney transplant recipients (**Diabetic KTR**)^{1,2}

Importantly, our subgroup analysis revealed that in patients with a diabetes diagnosis before the index date, the use of SGLT-2i could significantly reduce all-cause mortality, MACE, and MAKE, as demonstrated in our main analysis (**Figure 3**). We therefore change the title into: "The Outcomes of SGLT-2 Inhibitor Utilization in Diabetic Kidney Transplant Recipients" which encompasses both those with pre-existing and post-transplant diabetes. This revision will ensure that our manuscript accurately communicates the scope of our research and the characteristics of the studied population.

Reference:

1. Lim JH, *et al.* The Efficacy and Safety of SGLT2 Inhibitor in Diabetic Kidney Transplant Recipients. *Transplantation* **106**, e404-e412 (2022).
2. Sánchez Fructuoso AI, *et al.* Sodium-glucose cotransporter-2 inhibitor therapy in kidney transplant patients with type 2 or post-transplant diabetes: an observational multicentre study. *Clinical kidney journal* **16**, 1022-1034 (2023).

2. Additionally, aligning the index date of the PTDM study with the date of kidney transplantation is incorrect. The index date should be adjusted to the date when PTDM was diagnosed.

Response: Thank you for your insightful feedback regarding the alignment of the index date in our study. Our study's design intentionally aligns the index date with the date of kidney transplantation for all participants, including those with pre-transplant diabetes and those diagnosed with PTDM. Among the 1,995 SGLT-2i users in our study, 66 were diagnosed post-transplant compared to 2,218 among the 26,834 non-users. Most of the enrollee were

patients with T2DM before kidney transplantation. Aligning the index date with the transplantation date rather than the date of PTDM diagnosis allows us to standardize the follow-up period for all study participants. This approach reduces potential confounding factors that could arise if patients in the treatment group were followed from a variable start date depending on when they began medication. Such variability could introduce bias, making it challenging to match treatment and control groups accurately and fairly. To address your concerns, we have established a 90-day study selection period after kidney transplant for all participants. Additionally, we further conducted separate landmark analyses to ensure consistency across different selection periods (**Supplemental Table S3**). This approach showed similar results as our main analysis. We appreciate your understanding and continued support in improving our research.

3. If the study correctly addresses PTDM, the authors' assertion that more than 50% of the patients had diabetic nephropathy at the baseline index date (Table 1) is peculiar and needs clarification.

Response: Thank you for your comments regarding the prevalence of diabetic nephropathy at baseline. This high incidence is due to our cohort including many patients with type 2 diabetes prior to transplantation, alongside those diagnosed with PTDM. Diabetic nephropathy is a common complication in such patients, which explains its significant presence in our study group. We have updated the manuscript to better clarify the composition of our cohort and the extent of pre-existing diabetic complications. According to your comments, we have further revised the term to diabetic kidney transplant recipients (**Diabetic KTR**)^{1,2} and refined the title for more precision as “The Outcomes of SGLT-2 Inhibitor Utilization in Diabetic Kidney Transplant Recipients”. This revision will provide clearer insights into the clinical background of the study participants, aligning with the objectives of our research. We appreciate your input, which has significantly contributed to

refining our study's presentation.

Reference:

1. Lim JH, *et al.* The Efficacy and Safety of SGLT2 Inhibitor in Diabetic Kidney Transplant Recipients. *Transplantation* **106**, e404-e412 (2022).
2. Sánchez Fructuoso AI, *et al.* Sodium-glucose cotransporter-2 inhibitor therapy in kidney transplant patients with type 2 or post-transplant diabetes: an observational multicentre study. *Clinical kidney journal* **16**, 1022-1034 (2023).

4. The authors should demonstrate the differential effects of SGLT-2 inhibitor use between patients with diabetes before transplantation and those who developed diabetes post-transplantation.

Response: Thank you for your valuable suggestion to analyze the differential effects of SGLT-2 inhibitor use between patients with diabetes before transplantation and those who developed diabetes post-transplantation. We appreciate your attention to the distinction between PTDM and Type 2 diabetes. In our study, we included all patients with a diabetes diagnosis, whether pre-existing or developed post-transplant. Our subgroup analysis revealed that in patients with a pre-existing diabetes diagnosis before the index date, the use of SGLT-2 inhibitors also significantly reduced all-cause mortality (HR 0.33, 95% CI 0.23-0.47), MACE (HR 0.49, 95% CI 0.38-0.64), and MAKE (HR 0.54, 95% CI 0.45-0.66), consistent with our main analysis (**Figure 3**). Thank you again for your suggestion, which has helped us refine and improve our analysis.

5. The actual extent of SGLT-2 inhibitor usage, how long after transplantation they were used, and the possibility of the effects being attributable to other medications like metformin need to be explored. Typically, SGLT-2 inhibitors are used in patients with good renal function, so it's crucial to assess if the patients had inherently good prognoses at the time of drug initiation or PTDM diagnosis, not just at transplantation.

Response: Thank you for your inquiry regarding the extent and duration of SGLT-2 inhibitor usage post-transplantation and the influence of other medications like metformin. Although we did not have the exact dates of SGLT-2 inhibitor usage during the 90-day period after the kidney transplant—a limitation acknowledged in the Strengths and Limitations section (Line 348-350)—we conducted a further **landmark analysis**. The usage of SGLT-2 inhibitors within 2, 6, and 12 months post-transplantation all showed significant improvement in all-cause mortality, MACE, and MAKE (**Supplemental Table S3**). Furthermore, our specificity analysis across different baseline kidney function categories (eGFR <15, 15-29, >30 ml/min/1.73m²) yielded consistent results with our main analysis (**Figure 3**).

Additionally, we used propensity score matching to control for potential confounders, including the use of other medications such as metformin. This adjustment ensures that comparisons between patients on SGLT-2 inhibitors and non-users are well-balanced. This rigorous matching process is also evident in our subgroup analysis concerning other oral hypoglycemic agents (OHA), including biguanides, insulin, and sulfonylurea, as depicted in **Figure 3**. By combining this methodology with our statistical adjustments, we provide a thorough evaluation of the impacts of SGLT-2 inhibitors while accounting for other influential factors. Thank you for raising this important issue, and we hope this clarifies our approach.

6. There is a significant concern about immortal time bias—some patients might have longer follow-up but shorter duration of SGLT-2 inhibitor use, while others might have shorter follow-up but longer usage duration. It's also possible that some patients did not use SGLT-2 inhibitors at the start of follow-up. The study needs to clarify how it dealt with this issue, or otherwise, sufficient statistical adjustments are required.

Response: Thank you for your astute observation regarding the potential for immortal time bias in our study, which could arise from variations in the duration of follow-up and SGLT-2

inhibitor use among patients. We recognize the importance of addressing this issue to ensure the accuracy of our findings. We further conducted separate landmark analyses to ensure consistency across different selection periods to avoid immortal time bias. The usage of SGLT-2 inhibitors within 2, 4, and 6 weeks post-transplantation all showed significant improvements in all-cause mortality, MACE, and MAKE (**Supplemental Table S3**). The landmark method has been shown to effectively correct for immortal person-time bias and provides unbiased estimates, particularly when there is no treatment effect^{1,2}.

To further align with your feedback, we conducted subgroup analyses based on whether patients were prior users before kidney transplantation or new users post-transplantation (**Figure 3**). This approach helps safeguard against bias resulting from differences in the timing of treatment initiation. We standardized the start of follow-up for all patients by uniformly setting the index date as the transplantation date and defining SGLT-2 inhibitor use as occurring within 1-3 months post-transplant. Our statistical analyses reveal that long-term SGLT-2 inhibitor users (those who used SGLT-2 inhibitors both before and after the index date) had significant improvements in all-cause mortality (HR 0.34, 95% CI 0.23-0.49), MACE (HR 0.58, 95% CI 0.44-0.76), and MAKE (HR 0.50, 95% CI 0.42-0.61). In contrast, new SGLT-2 inhibitor users, though showing less pronounced benefits, still observed significant improvements in MAKE (HR 0.46, 95% CI 0.27-0.79). This suggests that long-term use of SGLT-2 inhibitors offers greater benefits. Additionally, we conducted a dosage assessment and found that regardless of dosage level, SGLT-2 inhibitor use did not significantly alter our tracking results for all-cause mortality, MACE, or MAKE (**Figure 3**). Thus, any dosage level of SGLT-2 inhibitor usage could positively contribute to both primary and secondary outcomes, consistent with the findings of the original EMPA kidney study³. These subgroup analysis results have been added to **Figure 3** and discussed further in the manuscript's Results and Discussion sections (Line 221-224, 242-245, 300-303). Thank you for your valuable insights, which have significantly improved the quality of our analysis.

Reference:

1. Mi X, Hammill BG, Curtis LH, Lai EC, Setoguchi S. Use of the landmark method to address immortal person-time bias in comparative effectiveness research: a simulation study. *Statistics in medicine* **35**, 4824-4836 (2016).
2. Cho IS, *et al.* Statistical methods for elimination of guarantee-time bias in cohort studies: a simulation study. *BMC medical research methodology* **17**, 126 (2017).
3. Wanner C, *et al.* Empagliflozin and Progression of Kidney Disease in Type 2 Diabetes. *The New England journal of medicine* **375**, 323-334 (2016).

Minor Criticisms:

1. It is important to assess the risks of urinary tract infections, euglycemic diabetic ketoacidosis, and acute kidney injury associated with SGLT-2 inhibitor use, if not already done.

Response: Thank you for highlighting the importance of assessing various risks associated with SGLT-2 inhibitor use. In our study, we indeed observed a significant increase in the risk of diabetic ketoacidosis (DKA), as shown in **Figure 2**. Following your suggestion, we also thoroughly examined the risks of urinary tract infections and acute kidney injury associated with SGLT-2 inhibitor use. We found no significant differences in the rates of overall genitourinary infections, UTI, or urogenital candidiasis among patients. Specifically, the adjusted hazard ratios (aHR) with 95% confidence intervals (CI) were as follows: for any genitourinary infection (UTI + candidiasis), aHR 1.03 (95% CI 0.83-1.28); for UTI alone, aHR 1.02 (95% CI 0.82-1.27); and for urogenital candidiasis, aHR 1.30 (95% CI 0.74-2.28). Regarding acute kidney injury (AKI), the aHR was 0.95 (95% CI 0.85-1.07), indicating no increased risk associated with SGLT-2i use. These findings have been added to **Figure 2** and discussed in our Results and Discussion sections (Line 187-192, 318-320). We appreciate your advice as it has guided a more comprehensive analysis of the associated risks.

2. The term "renal" should be replaced with "kidney" throughout the document, i.e., "renal

transplantation" should be "kidney transplantation", and "renal replacement therapy" should be "kidney replacement therapy".

Response: Thank you for your guidance on the use of terminology. We have revised the manuscript to replace all instances of "renal" with "kidney". This change is now consistent throughout the document, including terms like "kidney transplantation" and "kidney replacement therapy". We value your attention to detail which helps in maintaining consistency and clarity in our manuscript.

Reviewer #1:

I think the authors have thoughtfully taken on board reviewer comments and dealt with all queries and comments in a thorough manner. I think the manuscript is definitely much improved and ready for publication.

Response: Thank you very much for your positive feedback and for recognizing the efforts we have made in addressing the comments and queries. We are grateful for your support and are pleased that you find the manuscript much improved and ready for publication.

Reviewer #2:

Thank you for addressing my previous comments so thoroughly. I am satisfied with the comprehensive responses provided, and I appreciate the meticulous attention given to my feedback.

I particularly commend the authors for incorporating landmark analyses, which is an excellent idea to enhance the robustness of the findings. As a final recommendation, I suggest that the authors update the main table to include the landmark analyses results. Additionally, it would be beneficial to describe the landmark analysis methodology in the Statistical Analysis section of the Methods part of the manuscript.

Thank you for your diligent work on these revisions.

Response: Thank you for your constructive comments and for acknowledging our thorough responses.

In response to your further recommendations:

1. We have updated the main table to include the results of the landmark analyses in **Table 2** as suggested (Landmark Analysis of Outcomes of Interest at Multiple Temporal Points Post-Transplant: Comparison of SGLT-2i Users Versus Non-Users).
2. We have also added a description of the landmark analysis methodology in the Methods part of the manuscript (Line 83-87, 136-137).

We believe these additions will further enhance the robustness and clarity of our findings. Thank you again for your valuable feedback and support throughout this revision process.

Reviewer #3:

1. The authors should employ a new-user design to avoid selection bias (including individuals who were on SGLT2-is pre-transplant to those who initiated post-transplant).

Response: Thank you for your insightful comment regarding the potential for selection bias and the suggestion to employ a new-user design.

In our initial subgroup analysis (**Figure 3**), we have distinguished between individuals who were on SGLT-2i pre-transplant (long-term users) and those who were initiated with SGLT-2i post-transplant (new users). Because of limited patients, our findings indicated that there was no significant difference in mortality and MACE (Major Adverse Cardiovascular Events), while MAKE (Major Adverse Kidney Events) showed significant improvement for the new user group. For long-term users, all three outcomes (mortality, MACE, and MAKE) were significantly improved.

It is important to note that, historically, there has been little guidance on the use of SGLT-2i post-transplant for the DM management among kidney transplant recipients. Consequently, the number of patients initiating SGLT-2i post-transplant might be relatively low, and a new-user design did not yield sufficient statistical power. This limitation is consistent with real-world clinical practice and guidelines, which restrict the prescription of SGLT2i to a small subset of patients¹. In response to your query, Bayesian analysis can be particularly effective for small studies with good statistical power^{2, 3}. Therefore, we attempted to conduct a Bayesian analysis to assess the impact of SGLT2i treatment on patient mortality. The prior probability ($P(A)$) that a patient received SGLT2i treatment was 0.5. The likelihood ($P(B|A)$) of death among treated patients was 0.0208; while the baseline mortality rate ($P(B|A')$) among untreated patients was 0.0954. The overall probability of mortality ($P(B)$) in the patient population was 0.0581. The posterior probability ($P(A|B)$) that a deceased patient had received SGLT-2i treatment was 0.1790. The benefit probability, indicating the reduction in mortality due to the SGLT-2i treatment, was 0.0836. These findings imply that SGLT2i treatment could reduce mortality by approximately 8.36%. Nonetheless, we acknowledge the value of a new-user design in minimizing bias, as suggested by the reviewer, and will consider integrating such approach in future studies when feasible. We appreciate your suggestion and its contribution to enhancing the robustness of our research design.

Reference:

1. Vivarelli M, *et al.* The role of complement in kidney disease: conclusions from a Kidney Disease: Improving Global Outcomes (KDIGO) Controversies Conference. *Kidney international*, (2024).
2. Kelter R. Bayesian Hodges-Lehmann tests for statistical equivalence in the two-sample setting: Power analysis, type I error rates and equivalence boundary selection in biomedical research. *BMC medical research methodology* **21**, 171 (2021).
3. Williams J, Ferreira MAR, Ji T. BICOSS: Bayesian iterative conditional stochastic search for GWAS. *BMC bioinformatics* **23**, 475 (2022).

2. The authors should treat the exposure (initiation of an SGLT2) as a time-varying exposure. This will allow them to include individuals who die or use dialysis in that 3-month period. This can be achieved using the clone-censor-weight approach (<https://doi.org/10.1136/bmj.k182>). This will also allow the authors to use all the data available to them, instead of discarding the majority of the individuals who do not initiate an SGLT2-i as they have in their propensity-score matched analysis.

Response: Thank you for your insightful suggestion regarding the use of a time-varying exposure and the clone-censor-weight approach. We understand the importance of this method for capturing the dynamic nature of treatment initiation and its potential impact on outcomes.

Given the limitations of our current data source, which is aggregate data rather than individual-level data from TrinetX (Line 55-56), we were unable to implement the clone-censor-weight approach as suggested. However, we have conducted alternative analyses to address your concerns and to maximize the use of our data.

In response to your comments, we performed an extended analysis to examine the impact of outcomes between two groups of KTR: those who used SGLT-2i during the first 3 months post-transplant and continued usage from 3 to 6 months, and those who did not use SGLT-2i during the first 3 months and continued non-usage from 3 to 6 months (**Supplemental Table S5**). The adjusted hazard ratio (aHR) for all-cause mortality among SGLT-2i users was 0.47 (95% CI: 0.30 - 0.73, $p < 0.01$) compared to non-users. For MACE, the aHR for SGLT-2i users was 0.66 (95% CI: 0.49 - 0.90, $p < 0.01$). Similarly, for MAKE, the aHR for SGLT-2i users was 0.56 (95% CI: 0.45 - 0.70, $p < 0.01$). These results indicate significant reductions in all-

cause mortality, MACE, and MAKE for diabetic kidney transplant patients who continued using SGLT-2i beyond the initial 3-month period. Patients were categorized based on the presence of medication codes during these time periods to determine continued usage or non-usage of SGLT-2i (Line 186-194). Additionally, our original landmark analysis also showed that if we set the landmark time at 6 months or 1 year, our results remain significant (**Table 2**). Landmark analysis, which involved initiating the follow-up period at different time-frames, also showed the persistent benefit of SGLT-2i, even with intermittent use. This ensures the robustness of our results with ITT study design, mitigating the possibility of guarantee-time bias or immortal time bias. To further assess the comprehensiveness of our study, we conducted long-term risk analyses at 1 year, 3 years, and 5 years post-transplant (**Supplemental Table S3**). These results demonstrated again significant differences in all-cause mortality between SGLT-2i users and non-users across various time points (Line 183-186). These extended analyses confirm the robustness of our findings, demonstrating that the beneficial effects of SGLT-2i usage in reducing all-cause mortality, MACE, and MAKE are consistent across different time periods post-transplant.

Finally, it is pertinent to acknowledge that our analysis is based on aggregate data rather than individual-level data, which constrains our ability to utilize more sophisticated methodologies such as randomized controlled trial emulation, prescription time-distribution matching, and time-varying exposure models, including the clone-censor-weight approach. This limitation has been duly noted in the revised manuscript (Line 362-368).

We appreciate your understanding and the constructive feedback provided, which has significantly contributed to enhancing the rigor and validity of our study

3. Hazard ratios are flawed measures (10.1097/EDE.0b013e3181c1ea43). It would be prudent of the authors to present absolute risks (and risk differences), to let readers decide for themselves if there is a clinically meaningful effect. There may be in this scenario, as kidney transplant recipients have a high burden of comorbidities. However, when publishing results that suggest such a big reduction in mortality, the authors would be wise to be more thorough in their analysis strategies.

Response: Thank you for your valuable suggestion regarding the use of absolute risk and risk difference. We acknowledge that hazard ratios alone may not fully convey the clinical significance of our findings that suggest a big reduction in mortality, especially given the high burden of comorbidities in kidney transplant recipients.

In response to your comment, we would provide a new **Supplemental Table S3** that includes the absolute risk, risk difference, odds ratio, and risk ratio for mortality. Our analysis shows that SGLT-2i users had an absolute risk of 2.08% for five-year mortality, compared to 9.54% for non-users. This corresponds to a risk difference of -7.46% (95% CI: -8.90% to -6.02%), indicating a significant reduction in absolute risk. The risk ratio was 0.22 (95% CI: 0.16 to 0.30), and the odds ratio was 0.20 (95% CI: 0.14 to 0.28) (Line 165-170). We believe that this additional information would offer a more comprehensive perspective on the clinical impact of our findings and assist in informed decision-making in clinical practice.

We appreciate your recommendation and are committed to ensuring that our analysis is thorough and meaningful for our readers.

4. While including a negative exposure (Vitamin D) as a control is a good idea, the one

selected by the authors is unlikely to have a similar confounding structure to that of the original treatment strategy (SGLT2-i). Individuals choose to initiate Vitamin D for many different reasons than the reasons they get prescribed SGLT2-is.

Response: Thank you for your insightful comment regarding the choice of a negative control. We fully agree with your observation that Vitamin D might be prescribed for various reasons in patients with kidney disease, which could introduce confounding factors.

In our study, we did not use Vitamin D as our negative control. Instead, we selected Vitamin C as the negative control in our study design (**Figure 2**). We believe that using Vitamin C, which is less likely to be influenced by the same confounding factors as SGLT-2i, will help avoid potential controversies and provide a more appropriate comparison.

We appreciate your suggestion and are committed to ensuring the robustness and validity of our study.

5. The matching (or outcome adjustment, if the authors choose to alter their analysis strategy) would benefit from including pre-transplant eGFR and pre-transplant blood pressure (SBP and DBP, or MAP), both of which have an effect on treatment choice and mortality. These may have a large association with SGLT2-i initiation and mortality, violating the e-value study the authors have included in Supplemental information.

Response: Thank you for your detailed and thoughtful suggestion. We agree that pre-transplant eGFR and pre-transplant blood pressure (SBP, DBP) are important variables that can influence treatment choice and outcomes.

In our propensity score matching (PSM) analysis (**Table 1**), we have indeed included pre-

transplant eGFR, SBP, and DBP. We provided this information both before and after matching. After PSM, the standardized differences for eGFR, SBP, and DBP were 0.02, 0.01, and 0.04, respectively, indicating that the two groups were well balanced on these covariates.

We appreciate your recognition of the importance of these factors and will ensure that their inclusion and relevance are clearly highlighted in our manuscript. Your input is invaluable in refining our analysis and improving the robustness of our findings.

Reviewer #4:

Response: Thank you for your role in co-reviewing our manuscript as part of the Nature Communications initiative. We appreciate the feedback provided and the opportunity for Early Career Researchers to be recognized and trained in the peer review process. Your efforts and contributions to the review process are highly valued.

Reviewer #3:

1. Please add the new user analysis to the supplemental material. Even if it is not sufficiently powered, it is still interesting to see if the point estimates are in the same direction, and whether it is of a similar magnitude, to the point estimates in the main analysis.

Response: Thank you for your valuable feedback.

We have included the Methods for the New-User Analysis in the supplementary material, as requested. While the analysis may not be sufficiently powered, we agree that it is valuable to compare the point estimates with those from the main analysis to observe whether they are in the same direction and of similar magnitude.

2. We are generally concerned about the use of aggregated data to conduct comparative effectiveness research. That said, we recognize that there are not other databases available with this level of detail, and some evidence is better than no evidence. Please add this to the limitations section of the manuscript.

Response: Thank you for your insightful feedback.

We understand your concerns regarding the use of aggregated data for conducting comparative effectiveness research. We agree that while this presents certain limitations, the detailed data available from the TriNetX platform offers valuable insights, even if individual-level data is not accessible. As per your suggestion, we have added this point to the limitations section (Line 310-313 in main text) of the manuscript to acknowledge the trade-off between the limitations of aggregated data and the benefit of having detailed information.

Reviewer #4:

Response: Thank you for your information and your role in co-reviewing our manuscript as part of the *Nature Communications* initiative. We appreciate the feedback provided and the opportunity for Early Career Researchers to be recognized and trained in such an organized way of the peer review process. Your efforts and contributions to the review process are highly appreciated.